# Assessing Ecological Disturbance in Neotropical Forest Landscapes Using High-Level Diversity and High-Level Functionality: Surprising Outcomes from a Case Study with Spider Assemblages

**Darinka Costa Gonzalez** [1,2,*] **, Reinaldo Lucas Cajaiba** [1,3] **, Eduardo Périco** [4] **, Wully Barreto da Silva** [5] **, Antônio Domingos Brescovite** [6] **, António Maria Luis Crespi** [5] **and Mário Santos** [1,3]

1   Laboratory of Applied Ecology, CITAB—Centre for the Research and Technology of Agro-Environment and Biological Sciences, Institute for Innovation, Capacity Building and Sustainability of Agri-Food Production (Inov4Agro), University of Trás-os-Montes e Alto Douro, 5000-911 Vila Real, Portugal; reinaldo.cajaiba@ifma.edu.br (R.L.C.); mgsantos@utad.pt (M.S.)
2   Campus da Auga, University of Vigo, 32004 Ourense, Spain
3   Centre for Research and Technology of Agro-Environmental and Biological Sciences (CITAB), Inov4Agro, University of Trás-os-Montes e Alto Douro, 5000-801 Vila Real, Portugal
4   Laboratory of Ecology and Evolution, University of Taquari Valley, Lajeado 95900-000, Brazil; perico@univates.br
5   Laboratory of Ecology and Conservation, Federal Institute of Education, Science and Technology of Maranhão, Buriticupu 65393-000, Brazil; wully.silva@ifpa.edu.br (W.B.d.S.); acrespi@utad.pt (A.M.L.C.)
6   Laboratory of Zoological Collections, Butantan Institute, São Paulo 05503-900, Brazil; brescovit@butantan.gov.br
*   Correspondence: darinkacostagonzalez@gmail.com; Tel.: +351-259-350-238; Fax: +351-259-350-480

**Abstract:** Spiders have been increasingly used as environmental and ecological indicators in conservation and ecosystem management. In the Neotropics, there is a shortage of information regarding spiders' taxonomies and ecological responses to anthropogenic disturbances. To unravel these hitches, we tested the possibility of using high-level diversity and high-level functionality indicators to evaluate spider assemblages' sensitivity to landscape changes. This approach, if proven informative, might overcome the relevant limitations of taxonomic derived indexes, which are considered time-consuming, cost-demanding and dependent on the (few) expert taxonomists' availability. Our results highlight the pertinence of both indicators' responses to the structural changes induced by increasing anthropogenic disturbance, and are associated with reductions in ecosystem complexity, microclimates, and microhabitats. Overall, both indicators were sensitive to structural changes induced by anthropogenic disturbance and should be considered a useful resource for assessing the extent of ecosystems' disruptions in the Neotropics, and also to guide managers in landscapes' restoration.

**Keywords:** Neotropical forests; spider's assemblages; ecological indicators; high-level functionality; high-level diversity; anthropogenic disturbance

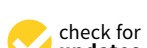



## 1. Introduction

Neotropical forest landscapes include some of the most threatened and unique ecosystems in the world, many have been reduced to small fragments dispersed within their historical distribution [1]. Even though recognized for the fundamental ecosystem services (ESs) they deliver, namely biological conservation, climate regulation, carbon sequestration, among others, they are subject to a snowballing pressure emerging from the interplay between the resources needed for a regional growing population, global market demands and climate teleconnections [2–5].

Changes in Neotropical ecosystems are precipitated by large-scale deforestation (partially illegal), wildfires with increased periodicity and extension (mostly human-sparked,

aggravated by the rising occurrence of extreme climatic conditions), selective logging, and the capture of high value species, all ultimately promoted by a fast-growing human population triggering upsurges in urban, intensive agriculture and cattle ranching areas [6,7]. The novel anthropogenic landscape composition and structure and land use intensity jeopardizes the provision of important ecosystem services and aggravates biodiversity erosion, namely of endangered species and keystone species [7,8].

Biodiversity, in all its dimensions (e.g., genetic, taxonomic and functional diversity), is considered the main provider of ESs [2,9–11], and is being particularly jeopardized through Neotropical ecosystems' contraction and isolation, including of taxa still unknown (or undescribed) for science [12]. Additionally, several works highlight significant changes in the ecological patterns and processes, triggering irreversible tipping points, at both functional and structural levels [2,9,10,13].

Understanding the sources of disturbances and the consequences of multiple interacting drivers shaping diversity and functional patterns is considered of utmost importance in order to define assertive policy measures and support pertinent management actions [7,14,15].

Ecological indicators, used to estimate consequences of diverse anthropogenic disturbances on natural communities, are considered a powerful decision-support tool, as their premature signaling might help in guiding the best management options to halt and even reverse undesirable trends [16–18]. Arthropods, through their functional and taxonomic diversity, play central roles in ecosystems [19]. Moreover, their sensibility to habitat changes, rapid responses to disturbances, and easy and cost-effective sampling demonstrates their indicator potential [20,21].

Within arthropods, predators such as spiders are particularly informative by, generally, occupying the top trophic levels and, in this way, are expected to integrate the biotic and abiotic influences affecting lower trophic levels [22,23]. In addition, they occur worldwide, are incredibly biodiverse (over 50,000 known species within 100,000 estimated to exist) [24–26] and prey on a diverse range of species using different foraging strategies [24]. In fact, they have been used as surrogates of conservation value and biodiversity, but mostly to detect the effects of land use disturbance and the restoration of ecosystems [27–31]. When grouped within guilds, they have been found to respond to microhabitat patchworks, including stubble changes in vegetation complexity [29,31,32]. However, spider surveys are particularly time consuming and require taxonomists with knowledge on the regional biota being studied. Overcoming these limitations is of great importance for detecting potential effects of fragmentation and anthropogenic-derived dynamics in Neotropical forest landscapes, and to support managers on decision-making processes [16,33,34].

The objectives of the present work were to estimate the usefulness of high-level diversity and high-level functionality by blending recognizable taxonomic units and recognizable functional units, respectively, in detecting disturbance patterns within a gradient of land use change in an Amazon forest landscape [35,36].

We were particularly interested in responding to specific questions, explicitly: do land use disturbance gradients correlate with high-level diversity and high-level functionality shifts? If correlated, is the response of these indexes informative, i.e., easy to holistically understand and explain? If the responses to the previous questions are positive, is it possible to suggest simple guidelines/framework on using high-level diversity and high-level functionality as complementary indicators to help in landscape and conservation management actions in the Neotropics?

## 2. Materials and Methods

### 2.1. Study Sites

The studied area was located in the municipality of Uruará, Pará, northeastern Brazil (Figure 1). The region is (still) dominated by old-growth primary forests, even though high deforestation rates, associated with livestock production and the exploitation of timber, especially in the south-central of the state and near the main roads, are originating a

novel agricultural landscape [12,37]. Studied sites were selected by taking into account the relevance of acquiring data from the dominant ecosystems within increasing levels of anthropogenic disturbance, categorized as Primary Forest (PF)—mostly old-growth forest, Secondary Forest (SF)—secondary vegetation with 15 years of regeneration, Incipient forest (IF)—secondary vegetation with 5 five years of regeneration, Agroforestry (AF)—cocoa plantations, and Pasture (PA)—extensive pastures for livestock [37] (Figure 1, complementary information on the ecosystems sampled is depicted in Tables S1 and S2 in Supplementary Materials).

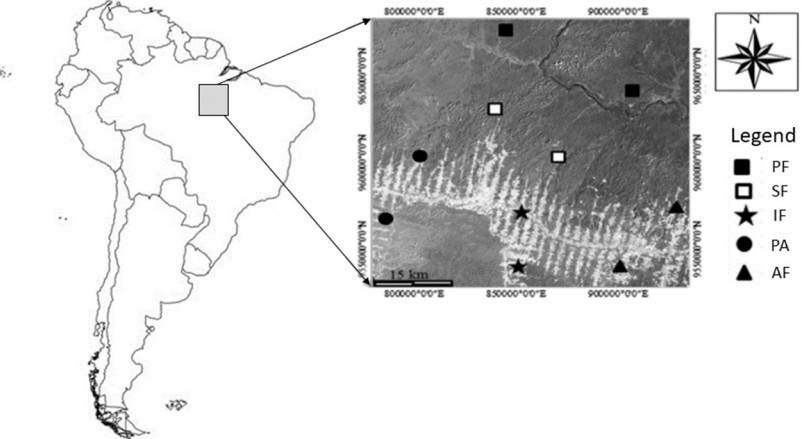

**Figure 1.** Study region of municipality of Uruará, state Pará, northern Brazil. Location of the ecosystem sampling sites: Primary Forest (PF), Secondary Forest (SF), Incipient succession (IF), Agroforestry (AF)—cocoa plantations, and Pasture for extensive livestock (PA).

*2.2. Data Collection and Identification*

Sampling was carried out during the year 2015, during the rainy season (February/March), mid-season (June) and dry season (September/October), to capture possible seasonal discrepancies in spider assemblages. Roofed pitfalls of 75 mm diameter and 110 mm deep were filled with preservative liquid made of water and neutral surfactant to brake the water tension and coarse salt to conserve individuals captured. Within each ecosystem, three independent areas (minimum distance of 5 km between areas) were chosen and five pitfall traps (minimum distance of 150 m between pitfalls) were placed for 48 h during the three seasons monitored (dry, intermediate, and rainy seasons) [9].

Pitfall traps were placed randomly whenever the conditions on the ground where favorable (i.e., no physical or biological barriers such as rocks, trees, etc.), always with a minimum distance of 100 m from ecotones, to ensure that most individuals captured were associated with the ecosystem being monitored.

Even if most captures were associated with soil surface dwelling spiders, pitfalls were considered a harmonizing method to be implemented while encompassing similar biases for all ecosystems considered [38]. In addition, the sampling period and intensity, totaling 225 traps, was outlined to link assemblage patterns with ongoing changes, without hampering a full community description [12,39,40]. Thus, we considered each pitfall trap as the unit base of our database used to carry the analysis. Taking into consideration the possibility of using our method for other regions and case studies, individuals were identified up to high-level recognizable taxonomic units (within families, hereon RTUs) and high-level recognizable functional units (within guilds, heron RFUs) for evaluating diversity or functionality, respectively [41–44], using specialized keys, reference information, as well as experts' opinions (Table S3 in Supplementary material) [13,45,46]. RTUs and RFUs were considered in order to solve major drawbacks in biodiversity assessments of megadiverse regions: the identification of taxa is often not fully acknowledged, considering not only the lack of specialists in Neotropical spiders' systematics and reduced funding, but also the ongoing addition of new taxa (species and even genus) [45]. Specimens were

deposited in the Zoological collection of the Biology Department of the University of Pará and in the Arachnological collection of Instituto Butantan, São Paulo, Brazil (IBSP, curator: A. D. Brescovit).

### 2.3. Environmental Factors and Variables

The different ecosystems, Primary forest (PF), Secondary forest (SF), Incipient forest (IF), Agroforestry (AF) and Pasture (PA), as well as the seasons (Dry (D), Intermediate (I), Rainy (R)) were considered factors with possible effects on high-level diversity (HLD) and high-level functionality (HLF) spatial and temporal dynamics. Moreover, fourteen environmental variables were measured in tandem with field works, complementing the previous nominal variables: temperature (T), humidity (H), precipitation (P), circumference at breast height (CBH), circumference at ankle height (CAH), canopy cover (CC), richness of plants (RP), abundance of plants (AP), richness of shrubs (RS), abundance of shrubs (AS), percentage of exposed soil (PES), percentage of green (vegetation) cover (GC), percentages of leaf litter cover (LLC), and height of leaf litter (HLL) [12]. Information concerning the methodology associated with each variable monitoring is depicted in Table S5 in Supplementary Material.

### 2.4. Sampling Completeness within Ecosystems

Rarefaction curves were gauged to evaluate whether the sampling effort was adequate to monitor the RTUs and RFUs per ecosystem, using PAST3.26 software [47].

### 2.5. Taxonomic and Functional Diversity

HLD was underpinned on RTU spider families' abundance (HLDa) and RTU spider families' richness (HLDr), while HLF was supported by RFUs sharing ecological traits (spider guilds) [48]. HLF guilds were assembled by applying a cluster analysis undertaken with the UPGMA linking algorithm, using the Sørensen similarity coefficient [13], supported by the traits: type of web and method of active hunting; prey range (either stenophagous or euryphagous); vertical stratification (ground or vegetation); circadian activity (diurnal or nocturnal) [13,49,50].

### 2.6. Assemblage Analysis

Taxonomic Diversity and Functional Diversity Indicators' Responses to Different Land Use

To discern the potential effects of environmental variables on HLD and HLF, generalized linear models (GLzM) were applied using Package 'RcmdrPlugin.EZR' [51]. Given the high overdispersion of data, models were associated with Quasi-Poisson distributions and log link function. HLD and HLF composition within ecosystems were compared by using Permutational Multivariate Analysis of Variance—PERMANOVA and Non-Metric Multidimensional Scaling (NMDS) analysis (using the Bray–Curtis index to HLF). Before the NMDS, environmental variables were fitted to the two first axes of ordination by the envfit function—in order to decrease bias associated with possible correlated explanatory variables, a nonparametric correlation was used beforehand (Spearman's rs < 0.75) [52]. This procedure selected the following environmental variables: temperature (T), humidity (H), percentage of leaf litter cover (LLC), height of leaf litter (HLL), and percentage of exposed soil (PES).

Beta diversity (β-diversity) partition analysis for HLD and HLF within ecosystems was also gauged, following the framework proposed by [53]: dissimilarities might occur by the replacement of some RTUs and RFUs by others (βsim) or by the formation of nested subgroups of more diverse communities (βnes). In this way, a pairwise dissimilarity index (βsør) was partitioned into two components: turnover (βsim) and nestedness (βnes): βsør = βsim + βnes. All analyses were performed using the functions beta.pair from betapart package [53] within R 3.2.4 program [54]. To complement the previous analysis, dissimilarities in HLD and HLF between ecosystems were quantified using simi-

larity percentages analysis (SIMPER), enabling the identification of the RTUs and RFUs accounting for most of the dissimilarities found.

Additionally, RTU and RFU indicators were identified using indicator value analysis (IndVal) [55] by combining specificity and fidelity within ecosystems and finding the significant values (R 3.2.4 program [54], using the Indic species package 1.7.5 [56] with 9999 permutations).

## 3. Results

### 3.1. Taxonomic and Functional Diversity Responses to Disturbance

A total of 809 individuals were collected, which were subsequently grouped within recognizable taxonomic units (RTUs) and recognizable functional units (RFU). Most individuals were captured in Agroforest (AF) sites (270 individuals, 18 RTUs and 6 RFUs) followed by Primary Forest (PF) sites (187 individuals, 19 RTUs and 7 RFUs), Secondary Forest (SF) sites (173 individuals, 19 RTUs and 7 RFUs), Incipient Forest (IF) sites (133 individuals, 16 RTUs and 8 RFUs) and Pasture (PA) sites (46 individuals, 15 RTUs and 9 RFUs) (Tables S3 and S4 in Supplementary Materials).

These individuals were grouped within taxonomic units (RTU): 647 in recognizable family abundance (HLDa), 32 in family richness (HLDr) and 9 in recognizable functional units (RFUs) (Table S2—Supplementary material). Most individuals were captured in Agroforest (AF) sites (270 individuals, 18 RTUs and 6 RFUs) followed by Primary Forest (PF) sites (187 individuals, 19 RTUs and 7 RFUs), Secondary Forest (SF) sites (173 individuals, 19 RTUs and 7 RFUs), Incipient Forest (IF) sites (133 individuals, 16 RTUs and 8 RFUs) and Pasture (PA) sites (46 individuals, 15 RTUs and 9 RFUs) (Tables S3 and S4 in Supplementary Materials). The most abundant RTUs were Segestriidae (130 individuals), Oonopidae (84 individuals), and Ctenidae (79 individuals) (Table S3 in Supplementary Material). Rarefaction curves did not reach their asymptote values (Figure 2a,b). However, this level of sampling completeness is likely to be sufficient to draw conclusions about the usefulness of HLD and HLF as indicators of anthropogenic impacts [57,58].

HLDa, HLDr and HLF were significantly influenced by specific ecosystems, seasons, and their interactions. The general trend depicted a reduction for all indexes within the studied disturbance gradient—from the more pristine ecosystems to the more disturbed ones—except for the extensively managed AF, with non-significant differences from the reference ecosystem (PF). On the other hand only, the I season had significant effects for HLDa, while interactions between ecosystems and seasons had complex and even antagonistic outcomes depending on the index (HLDa, HLD or HLF) (Figure 3 and Table S6 in Supplementary Material).

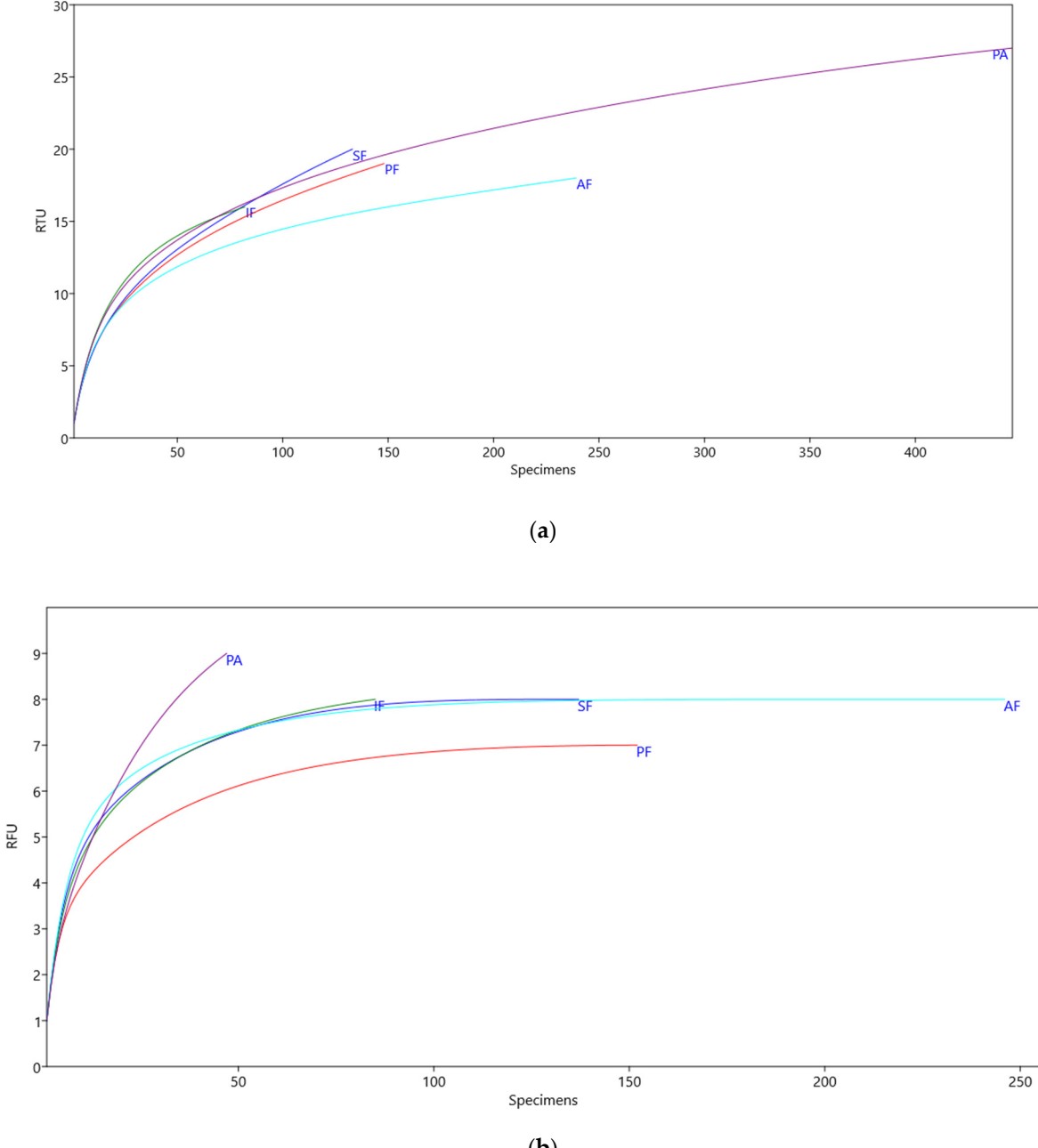

**Figure 2.** Individual-based rarefaction curves for the studied ecosystems: (**a**) high-level recognizable taxonomic units (RTU). (**b**) High-level recognizable functional units (RFU). Primary Forest (PF), Secondary Forest (SF), Incipient Forest (IF), Agroforestry (AF) and Pasture (PA).

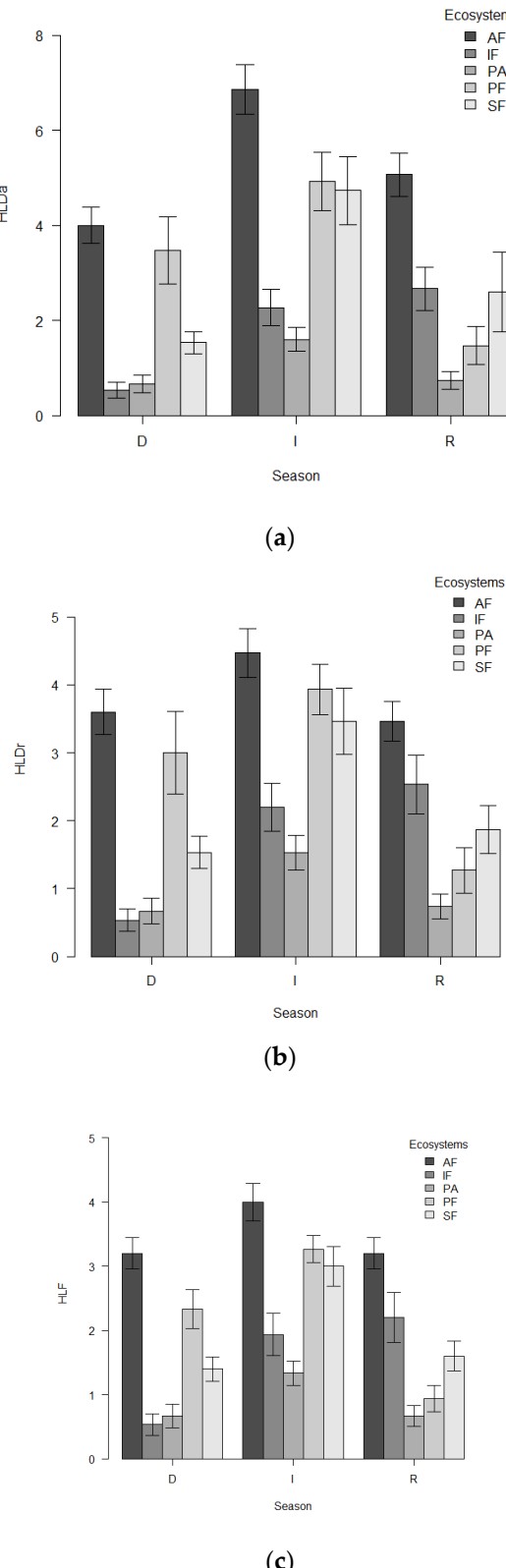

**Figure 3.** Bar plots expressing the mean differences in high-level diversity abundance (HLDa) (**a**), high-level diversity richness (HLDr) (**b**) and high-level functionality (HLF) (**c**), considering the interactions between the ecosystems and seasons considered. Ecosystems: Primary forest (PF), Secondary forest (SF), Incipient Forest (IF), Agroforestry (AF), and Pasture (PA); Seasons: Dry (D), Intermediate (I), Rainy (R). Standard Error with 95% confidence interval.

### 3.2. Taxonomic Composition and Functional Composition Responses to the Disturbance Gradient

Taxonomic (HLD) and functional (HLF) composition presented significant differences among the five ecosystems (PERMANOVA: F = 12.71, *p* < 0.0001 and F = 10.65, *p* < 0.0001, respectively) (Tables 1 and 2).

**Table 1.** PERMANOVA results (based on Bray–Curtis similarity) comparing the composition of high-level diversity richness (HLDr) assemblages between the ecosystems (F = 38.14, *p* < 0.0001).

| Ecosystems | PF | SF | IF | AF |
|:---:|:---:|:---:|:---:|:---:|
| PF | 0 | - | - | - |
| SF | 9.37 *** | 0 | - | - |
| IF | 7.32 *** | 11.88 *** | 0 | - |
| AF | 14.39 *** | 12.39 *** | 14.80 *** | 0 |
| PA | 18.16 *** | 17.51 *** | 15.67 *** | 7.96 *** |

Legend: Ecosystems: PF (Primary forest), SF (Secondary forest with 15 years of regeneration), IF (Incipient forest with five years of regeneration), AF (Agroforestry cocoa plantations), PA (Pasture). Significance: *** *p* < 0.0001.

**Table 2.** PERMANOVA results (based on Bray–Curtis similarity) comparing the composition of high-level functionality (HLF) assemblages between the ecosystems (F = 38.14, *p* < 0.0001).

| Ecosystems | PF | SF | IF | AF |
|:---:|:---:|:---:|:---:|:---:|
| PF | 0 | - | - | - |
| SF | 8.93 *** | 0 | - | - |
| IF | 4.34 *** | 9.78 *** | 0 | - |
| AF | 13.39 *** | 9.88 *** | 12.34 *** | 0 |
| PA | 12.46 *** | 15.21 *** | 2.54 * | 23.97 *** |

Legend: Ecosystems: PF (Primary forest), SF (Secondary forest with 15 years of regeneration), IF (Incipient forest with five years of regeneration), AF (Agroforestry cocoa plantations), PA (Pasture). Significance: * *p* < 0.01; *** *p* < 0.0001.

Non-Metric Multidimensional Scaling (NMDS) analysis of HLD complemented the previous results by depicting significant differences between PF assemblages and all other ecosystems, except for IF. Additionally, the remaining ecosystems (AF, SF, and PA) showed some degree of similarity (Figure 4a). NMDS first axis revealed positive correlations with humidity (H), height of leaf litter (HLL) and percentage of litter cover (LLC); the second axis revealed positive correlations with temperature (T), humidity (H), percentage of exposed soil (PES), height of leaf litter (HLL), and negative correlations were revealed with temperature (T) and percentage of litter cover (LLC) in the first and second axes, respectively (Table S7 in Supplementary Material). A correlation was found between PF and humidity (H), percentage of leaf litter cover (LLC) and height of leaf litter (HLL), while IF, AF, and PA appear to be more related to percentage of exposed soil (PES) and temperature (T). IF was mostly correlated with the height of leaf litter, and SF was related to humidity (H) and height of leaf litter (HLL) (Table S7 in Supplementary Material).

NMDS analysis of HLF also depicted significant differences between ecosystems: two groups were discriminated, encompassing PA and IF (associated with highly disturbed ecosystems) and SF and AF (associated with moderately disturbed ecosystems) when compared with PF (associated with pristine ecosystems) (Figure 4b). Additionally, AF and IF seem isolated from the other ecosystems (Table S8 in Supplementary Material). Concerning the environmental variables, temperature (T) and percentage of exposed soil (PES) were positively correlated with the first axis, while humidity (H), percentage of leaf litter cover (LLC) and height of leaf litter (HLL) were positively associated with the second axis (Table S8 in Supplementary Material).

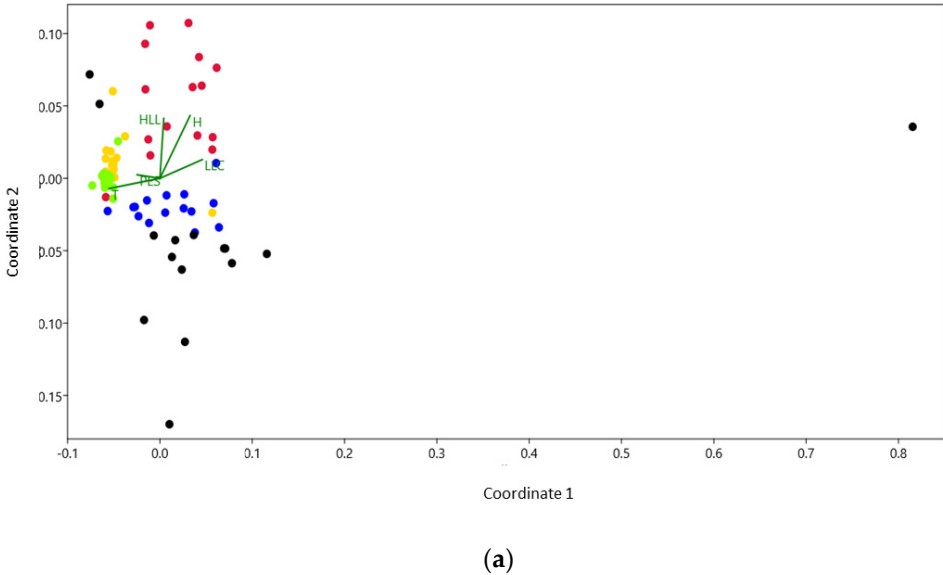

(**a**)

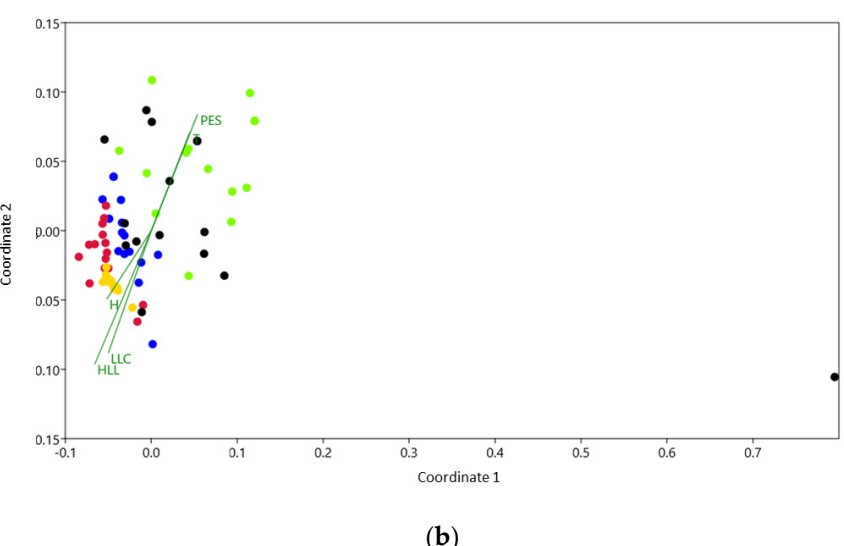

(**b**)

**Figure 4.** Non-metric multidimensional scaling (NMDS) using Bray–Curtis similarity; 5 environmental variables: (**a**) high-level diversity richness (HLDr) assemblages grouped in accordance with the ecosystems; (**b**) high-level functionality (HLF) assemblages grouped in accordance with the ecosystems. Legend: Ecosystems: PF (Primary forest) as blue, SF (Secondary forest succession with 15 years of regeneration) as red, IF (Incipient secondary succession forest with five years of regeneration) as black, AF (Agroforestry cocoa plantations) as yellow, PA (Pasture) as green. Environmental variables: T = Temperature; H = Humidity; PES = Percentage of exposed soil; LLC = Percentage of leaf litter cover; HLL = Height of leaf litter (cm).

### 3.3. Comparing Taxonomic β-Diversity Indicators with Functional β-Diversity Indicators

HLD beta diversity dissimilarity (Dβsor) presented relatively high values (Dβsor median = 0.33): PA vs. IF (Dβsor: 0.43), PA vs. PF (Dβsor: 0.42), PA vs. AF (Dβsor: 0.41), PA vs. SF (Dβsor: 0.39), PF vs. AF (Dβsor: 0.32), PF vs. SF (Dβsor: 0.30), SF vs. IF (Dβsor: 0.30), SF vs. AF (Dβsor: 0.29), IF vs. PF (Dβsor: 0.28) and IF vs. AF (Dβsor: 0.21) (Figure 5a). Turnover (Dβsim) was the dominant component of Dβsor: PA vs. IF (Dβsim: 0.41), PA vs. AF (Dβsim: 0.35) and PA vs. PF (Dβsim: 0.35), while IF vs. AF had the lowest values (Dβsim: 0.17) (Figure 5a). On the other hand, nestedness (Dβnes) ranked between SF vs.

PA (Dβnes: 0.09), SF vs. IF (Dβnes: 0.07), and PF vs. PA (Dβnes: 0.09) to PF vs. SF and PA vs. IF (Dβnes= 0.017 simultaneously) (Figure 5a and Table S9 in Supplementary Material).

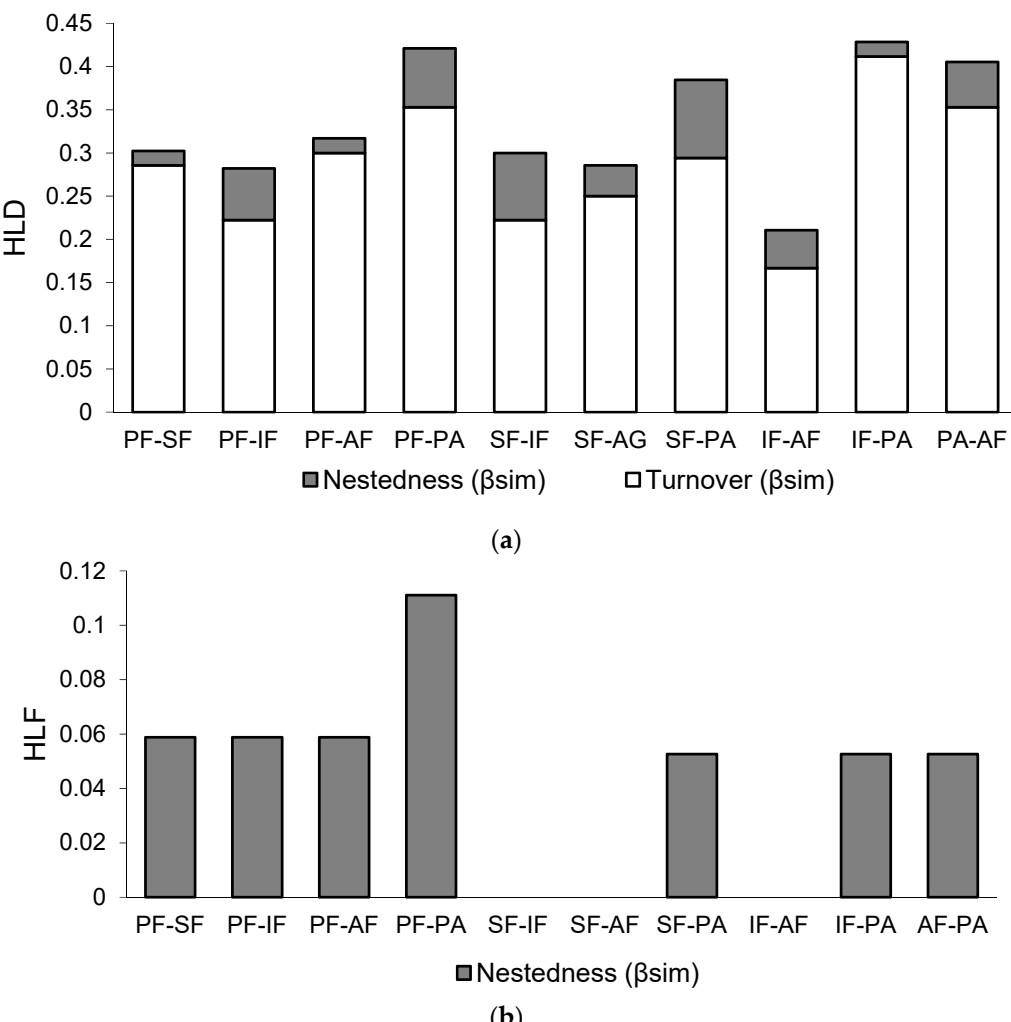

(**a**)

(**b**)

**Figure 5.** Spider assemblages' dissimilarities between ecosystems (Beta diversity (βsor), Turnover (βsim), and Nestedness (βnes)): (**a**) high-level diversity (HLD) dissimilarities; (**b**) high-level functionality (HLF) dissimilarities. Legend: Ecosystems: PF (Primary forest), SF (Secondary forest succession with 15 years of regeneration), IF (Incipient secondary succession forest with five years of regeneration), AF (Agroforestry cocoa plantations), PA (Pasture).

HLF beta diversity dissimilarity (Fβsor) was, in general, lower than Dβsor (Fβsor median = 0.04). Further, Fβsor was identical to Fβnes since Fsor presented nil results for all comparisons. Nestedness ranged from PA vs. PF (Fβsor/Fβnes = 0.11), followed by PF vs. SF, PF vs. IF, PF vs. AF, PA vs. SF, PA vs. IF, and PA vs. AF (Fβsor/Fβnes = 0.05). Fβsor/Fβnes associated to SF vs. IF, SF vs. AF, and IF vs. AF also presented no dissimilarities (Fβsor/Fβnes = 0) (Figure 5b and Table S10 in Supplementary Material).

### 3.4. Taxonomic and Functional Dissimilarity within the Disturbance Gradient

Ecosystem assemblages were also discriminated, using the SIMPER analysis, for both HLD and HLF. HLD dissimilarity depicted the RTUs of Salticidae (71%), Corinnidae (63%) and Lycosidae (60%) as the more discriminant within the ecosystems analysed (Table S8—Supplementary material). On the other hand, HLF dissimilarity was mainly related with orb web weavers (TA.OWW -(74%), other hunters (TA.OH -(67%), and sheet web weavers/other hunters (TA.ShWWOT -(67%) (Table S7 in Supplementary Material). Overall

percentage values for each ecosystem pairwise comparison and the main RTUs/RFUs that contributed to ecosystem dissimilarity are presented in Table S8 in Supplementary Material.

### 3.5. Families and Guilds as Disturbance Indicators

Indicator value analysis (Indval) [55] was able to discriminate RTUs and RFUs representing specific ecosystems within the disturbance gradient. Six RTUs were associated with AF (Segestriidae, Araneidae, Linyphiidae, Salticidae, Scytodidae, Palpimanidae), three with PA (Lycosidae, Paratropididae, Dictynidae), and one with PF (Ctenidae). Three RFUs were associated with AF (sheet web weavers/other hunters—TA.ShWWOT, space web weavers—TA.SWW, ground hunters—TA.OH), one to SF (orb web weavers—TA.OWW), one to IF (space web weavers—TA.SWW), and one to PA (space web weavers ground hunters—TA.SWWGH). (Table S12 in Supplementary Material).

## 4. Discussion

### 4.1. Taxonomic Diversity and Functional Diversity Responses to the Disturbance Gradient

Incomplete rarefaction curves detected for RTUs and RFUs are probably related with the relevant taxonomic and functional diversity encompassed by the ecosystems studied.

High-level diversity and high-level functionality (HLDa, HLDr and HLF) responded consistently to ecosystem substitution within the landscape studied. In fact, results pinpoint a tapering of abundance, richness, and functionality from the less disturbed ecosystems to the anthropogenic ones. Ecosystem structural simplification (e.g., vegetation structure, architectural complexity, and heterogeneity) might be partially responsible for reducing the niches' possibilities [58–66]. On the other hand, agroforestry's (AF) resemblance to primary forests (PF) confirms their potential to provide significant ecosystem services in the Neotropics (specially supporting services) [67–70]. In fact, the complex structure of AF, encompassing several micro-habitats, can support diverse recognizable taxonomic units (RTU) and recognizable functional units (RFU) [71–74]. This is especially relevant considering that cocoa agroforests are carved by management practices that stimulate the inception of diverse ecotones, microclimates, and microhabitats, which in turn influence the diversity of arthropods and spider hunting strategies [75]. In fact, certain RFU, such as dwelling spiders and web-building spiders, attain their highest diversity in AF [72].

Spider communities associated with IF seem separate from the other ecosystems, dominated by specific RTUs (the families *Dipluridae, Ochyroceratidae, Hahniidae, Pisauridae*) and RFU, namely "Sheet web weavers", characterized by nocturnal ground euryphagous spiders [13], probably conditioned by a contraction in arthropods' diversity [75,76]. Additionally, the lack of shade is associated with extreme daily (and seasonal) variation in microclimates, reducing the RTU suitability and RFU spectrum [77]. This trend is even more evident for pastures (PA), which depicted the lowest values of HLD and HLF, which might result from their continuous disturbance patterns (fire, cattle) and low complexity— dominated by herbaceous vegetation (mainly species of the genus *Brachiaria* spp.) used for extensive livestock farming (Table S1 in Supplementary Material) [71]. These are especially unfavorable conditions for RFU web spiders, which are associated with the reduced diversity of arthropods, hiding places, and microclimates, which disrupt mating behaviors as well as inter- and intraspecific competition [77–80].

### 4.2. Taxonomic Composition and Functional Composition Responses to the Disturbance Gradient

HLD Permanova/NMDS results associated with PF and IF assemblages' compositions depict similarities, as well as those of AF, SF and PA, which are apparently linked by structural complexity and the potentially derived ecological niches [65]. Even if similarities between IF and PF could be explained by the allocation of refuges for several RTUs [81], namely by IF mimicking PF's mosaic-like structure and heterogeneity [82,83]), we are deeply convinced that it might be connected to the biases of our method. Pitfall traps select mostly soil surface dwelling (epigean) arthropods, and therefore could lead to an under sampling of ecosystems characterized by stratification and higher canopy density

(such as SF and AF ecosystems [13,84,85]. Regardless, HLF Permanova/NMDS results enlighten the previous HLD trends, by depicting PF partially superimposed to all other ecosystems while clearly separating IF and PA from AG and SF. The more complex and heterogeneous ecosystems might be associated with RFUs with different requirements and diverse dispersal capabilities [65,86–88]. Thus, our results emphasize that while composition is scattered within the ecosystems under study, functionality shows a clearer separation between less and more disturbed ecosystems [13]. This supports the idea that high ecological redundancy might be fundamental to enhance ecosystem resilience in anthropogenic landscapes [13,89–91]).

*4.3. Taxonomic β-Diversity and Functional β-Diversity Responses to the Disturbance Gradient*

The dissimilarities of taxonomic β-diversity (Dβ-diversity) are mostly linked with the turnover (βsim) component, i.e., most differences between communities are related to a substitution of families. IF and PA, as well as PF and PA, are the ecosystems depicting higher dissimilarities. On the other hand, functional β-diversity (Fβ- diversity) dissimilarities are related exclusively with nestedness, expressing functional losses along the disturbance gradient. Further, Fβ- diversity presents reduced dissimilarities, probably arising from the smaller diversity of RFUs when compared with the diversity of RTU. No differences were detected in the comparisons of SF vs. IF and AF vs. SF, showing equivalent functionality in these intermediate disturbed systems.

Again, the results obtained might be directly linked with a gradient of structural complexity [13,92,93]; as an example, PA, the dissimilar ecosystem, is structurally simple and dominated by herbaceous vegetation (mainly species of the genus Brachiaria spp.) used for extensive livestock farming. Further, PF is composed by native RTUs and high structural complexity, housing a variety of physical conditions (soil and microclimate) that enforce the existence of microhabitats and thereby provide niches for diverse types of RFUs [92,94,95]. In fact, several studies indicate that biological diversity increases with structural diversity, which enhances niche availability [2,96,97], and structural complexity seems to be directly related to the diversity of other taxa [92,98–100].

*4.4. Taxonomic and Functional Dissimilarity within the Disturbance Gradient*

The results obtained throughout Indval/SIMPER analysis highlight the potential of using high-level diversity (e.g., RTUs) and high-level functionality (e.g., RFUs as ecological indicators of disturbance in Neotropical forest landscapes). Structure and environmental conditions play filtering processes by selecting RTUs and RFUs with physiological and behavioral adaptations [101,102]. In this way, pristine ecosystems were associated to RTUs (Cetenidae) that are dependent on a broad range of microhabitats, including the forest floor and tall vegetation [103]. Interestingly, agroforestry encompassed the largest set of RTU indicators (Segestriidae, Araneidae, Linyphiidae, Salticidae, Scytodidae, Palpimanidae) and RFU indicators (TA.ShWWOT, TA.SWW, TA.GH). In fact, cocoa agrogroforests are characterized by very complex structures, ranging from close canopy to open areas and tree architecture, offering a wide variation of ecological conditions [78,80]. Foliage diversity and density have determinant influences on spider bioecological behaviors such as web construction, prey availability, hiding places and microclimate conditions [78,80]. The thinning of cover reduces the opportunities for diverse predator strategies [102]. Interestingly, in PA, the RTU indicators (Lycosidae, Paratropididae, Dictynidae) and guild (TA.SWWGH) are characterized by being ground hunters and two main classes of behavior strategies: sit and wait predators (Paratropididae) and active hunters (Lycosidae) [102]. In addition, high dispersal abilities are linked to less complex ecosystems [65,86], such as the indicators found in PA and IF, while low dispersal abilities characterize PF and AF RTU and RFU indicators [78,102,103].

### 4.5. Methodological Advantages and Drawbacks

HLD and HLF depict indicator sensitivities for a preliminary assessment of Neotropical forest landscapes' statuses, enabling the prediction and projection of anthropogenic changes [104–106]. When compared with traditional ecological impact assessment studies, our approach significantly lessens time and costs associated [107–109]. In this way, our study reinforces the use of HLD and HLF as useful proxies in biodiversity studies and long-term monitoring in the Neotropics, especially in cases where taxonomist lack data [110]. Even though pitfall traps capture individuals belonging to ground hunters, they also capture other guilds that live in low vegetation and/or the ground, such as web specialists, sheet web weavers, web weavers, and other hunters/ambushers [82,111]. Additionally, the general results obtained are highly correlated with other field sampling methods for spiders [112]. Nevertheless, the results should be corroborated with additional information on spider surveys using complimentary techniques [113], namely, considering the diversity behaviors and ecosystem structure preferences (ground, litter layer, herbaceous, shrubs and low stratum) [13,114,115]. Additionally, arthropod data collected with pitfall could be linked with spider data and ecosystem characteristics for an inclusive view of disturbances and/or the recovery of ecosystems [86,112,116]. Seasonal variation and the effects of vegetation type and structure on spider communities, as well as our ignorance of what shapes diversity and functionality in the Neotropics should also be acknowledged in future works using HLD and HLF [76].

Our preliminary results demonstrate that high-level indicators reflect gradients of disturbance and, consequently, might be considered ecological indicators of the status of Neotropical forest landscapes. Regardless, it is likely that the activity of more elusive spiders and species from other strata was probably not detected by pitfalls. Monitoring was accomplished during limited periods, which might not be representative of the full activity patterns of several species. Moreover, we also believe that increasing the sampling effort and adding more locations should be considered in order to check the feasibility of using our high-level framework as a guideline to monitor the status of Neotropical landscapes.

### 5. Conclusions

High-level diversity and high-level functionality, as proposed in our work, responded insightfully to the different ecosystems, as spider families and spider guilds revealed mounting sensitivity to changes within the disturbance gradient in a Neotropical forest landscape. Differences within ecosystems are mostly related to turnover and functional composition trends specifying reductions in ecosystem resilience from the more pristine to highly disturbed. Agroforests present values of high-level diversity and high-level functionality equivalent to primary forests and might be considered (partial) surrogates of pristine ecosystems, even though they have structurally different compositions. Our results strongly support the use of high-level diversity and high-level functionality indicators as preliminary assessment indicators of disturbance, and highlight the importance of using agroforestry land management as a complementary strategy to halt the loss of biodiversity and services associated with these fast-changing landscapes.

**Supplementary Materials:** The following are available online at https://www.mdpi.com/article/10.3390/land10070758/s1, Table S1: Generic characteristics of the ecosystems sampled in the municipality of Uruará, state Pará, northern Brazil, Table S2: Environmental variables monitored during fieldworks, Table S3: Abundance (N) and Richness (S) associated with the Recognizable Functional Units (RFU) and Recognizable Taxonomic Units (RTU) (Cardoso et al., 2011) in the ecosystems monitored in the Brazilian Amazon, Table S4: Total number of individuals captured the mean and the standard deviation per ecosystem, Table S5: Specification of the methodology used for monitoring the environmental variables associated with the ecosystems studied, Table S6: Generalized linear models (GLzM) relating high-level diversity abundance (HLDa), high-level diversity richness (HLDr) and high-level functionality (HLF) with the different ecosystems, seasons and their interactions, Table S7: Non-metric Multidimensional Scaling (NMDS) results for high-level diversity (HLD): environmental variables' and ecosystems' associations with the principal axis (axis 1 and axis 2), Table S8: Non-

metric Multidimensional Scaling (NMDS) results for high-level functionality (HLF) environmental variables' and ecosystems' associations with the principal axis (axis 1 and axis 2), Table S9: Depicting high-level diversity (HLD) dissimilarities, Table S10: depicting high-level diversity (HLD) dissimilarities, Table S11: SIMPER results comparing the composition of high-level diversity (Recognizable Taxonomy Units (RFU)) and high-level functionality (Recognizable Functional Units (RFU)) between ecosystems, Table S12: Indicator Recognizable Taxonomic Units (RTU) and indicator Recognizable Functional Units (RFU), derived from the Indicator value analysis (Indval), associated with the spiders of the ecosystems studied.

**Author Contributions:** Conceptualization, D.C.G. and A.D.B.; Data curation, R.L.C., E.P., W.B.d.S. and A.D.B.; Formal analysis, D.C.G., R.L.C., E.P., W.B.d.S., A.M.L.C. and M.S.; Investigation, D.C.G. and R.L.C.; Methodology, M.S.; Supervision, M.S.; Writing—original draft, D.C.G.; Writing—review & editing, D.C.G., R.L.C., E.P., W.B.d.S., A.D.B., A.M.L.C. and M.S. All authors have read and agreed to the published version of the manuscript.

**Funding:** This research received no external funding.

**Institutional Review Board Statement:** Not applicable.

**Informed Consent Statement:** Not applicable.

**Data Availability Statement:** Not applicable.

**Acknowledgments:** This work is supported by National Funds by FCT—Portuguese Foundation for Science and Technology, under the project UIDB/04033/2020 and ADB grant PQ 303903/2019-8.

**Conflicts of Interest:** The authors declare no conflict of interest.

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
