# Peer review of "Assessing Ecological Disturbance in Neotropical Forest Landscapes Using High-Level Diversity and High-Level Functionality: Surprising Outcomes from a Case Study with Spider Assemblages"

_land, doi:10.3390/land10070758_

Round 1
Reviewer 1 Report
Please find the remarks to the manuscript in the attached file.

Author Response
Dear reviewer, before our replies and explanations of the changes made in order to fulfil your comments and suggestions, we want to express our gratitude for the feedbacks and inputs made to our previous version of the manuscript. We believe that the comments and suggestions enabled a significant improvement of our work. Anyway, if the referee fells that some of the changes and comments are incomplete and/or deserve more information, we will be willing to modify in accordance with additional suggestions.
We have separated below the referees’ comments, highlighted in bold from the authors’ responses, which are depicted in plain text. In addition, we have pasted the changes made in the manuscript and in the supplementary materials using “” and italics, whenever possible.
Response to reviewers’ comments
Reviewers' comments:
Reviewer #1:
- Introduction: In this chapter, the authors concentrated mostly on the role of spiders as indicators of habitat changes. It would be good to add here a few sentences concerning the ecological disturbances taking place in Neotropical ecosystems. I think it would be important for readers to know specific examples of such adverse ecosystem changes and their scale, thus the readers would also be more convinced of the necessity of conducting this type of analysis as presented in the manuscript.
Response 1:
We acknowledge the reviewer comments. We have changed the introduction to reply to the reviewer suggestions. We might add more text if the reviewer considers our changes incomplete. Please find in the following lines some improvements that we have made in the manuscript text:
Lines [48 – 56]:
“Changes in Neotropical ecosystems are precipitated by large-scale deforestation (partially illegal), wildfires with increased periodicity and extension (mostly human-sparked, aggravated by the rising occurrence of extreme climatic conditions) and selective logging and capture of high value species, ultimately promoted by a fast-growing human population triggering upsurges in urban, intensive agriculture and cattle ranching areas [6, 7]. The novel anthropogenic landscape composition and structure and land use intensity jeopardizes the provision of important ecosystem services and aggravates biodiversity erosion, namely of endangered species and keystone species [7, 8].”
Lines [63 – 66]:
“Understanding the sources of disturbance and the consequences of multiple interacting drivers shaping diversity and functional patterns its considered of utmost importance in order to define assertive policy measures and support pertinent management actions [7, 14, 15].
____________________________________________________________________
Reviewer 1#2:
- Material and methods. First of all, please add in this chapter the information what the sample in the data basis in your analyses was. Did you sum up spiders from all traps in each environment from all study periods and this was a sample? It would be good to know more details.
Response 2:
We are grateful for the comments addressed here, that were considered most useful. In fact, after carefully revising the reviewer’s comments, we realized the importance of adding more detail to the sampling approach and that criteria to build our database. We also found a small mistake in our explanation of the sampling protocol, with no implications in the analysis made and conclusions.
Our analyses were performed taking as sample the number of individuals collected by pitfall. For each ecosystem studied (PF, SF, IF, AF and PA), three independent areas were selected (minimum distance of 5km from each other) and five pitfall traps (located with a minimum distance of 150 meters from each other) were placed for 48 hours during the considered seasons (Dry, Intermediate and Rainy). In this way, 225 traps were monitored (225 samples = 5 traps x 3 areas x 3 seasons x 5 ecosystems), each one corresponding to a pitfall trap. Biological data was organized per species or morphospecies or genus (whenever it wasn´t possible to identify to species level). Nevertheless, and to test our high-level hypothesis, individuals were grouped within spider family abundance (HLDa), spider family richness (HLDr) and guilds abundance (HLF). The reason for choosing this approach was because we wanted to test metrics for that we believe useful in regions under sampled and with many undescribed (or poorly known and difficult to be identified) species. In fact, for a high number of individuals captured, taxonomic identification to the species level was particularly difficult (no guides and/or lack of regional taxonomists), but also considering the stochastic variation in species’ captures, linked with unknown variables that could introduce more “noise” to our analysis then clarification.
Also, during the revision of this issue, we realized that we mentioned a wrong number of traps used (480 pitfall traps), when in fact, and as explained above, it was 225. We are deeply sorry for this confusion, and we apologize for this inaccuracy. This confusion aroused because during the period of data collection, other arthropods were also being monitored in the region (e.g. ground beetles, ants, etc) but with a different protocol that include baited traps. We would like also to inform the referee that this mistake had no implications in the analysis made and results obtain. We have corrected in the text the total number of pitfall traps to the actual number. (225). Hence, taking in account reviewer comments we have made the following changes in the manuscript text (we will be willing to add more information if asked by the referee):
Lines [123 – 126]:
“Within each ecosystem, three independent areas (minimum distance of 5km between areas) were chosen and five pitfall traps (minimum distance of 150 meters between pitfalls) were placed for 48 hours during the three seasons monitored (dry, intermediate and rainy season).”
Lines [170 – 172]:
“HLD was underpinned on RTU spider families’ abundance (HLDa) and RTU spider families’ richness (HLDr) while HLF was supported on RFU sharing ecological traits (spider guilds).”
___________________________________________________________________________
Reviewer 1#3:
2.1 Study sites: Please, complete the description of study sites. In Supplementary Material in Table A1 there are characteristics of the study plots, but in the description of different types of forests, there is information only about the tree level. There are no details about the ground level vegetation and its structure. This is important, because pitfall traps were placed on the soil surface level.
Response 3:
We acknowledge the reviewer suggestion, that was considered of key importance for increasing the readability of our manuscript and for a “visualizing” the ecosystems monitored. We have added information related to the ground level vegetation and its structure in table A1 – Supplementary Material. We hope it fulfils the reviewer expectations.
Table A1. Generic characteristics of the ecosystems sampled in the municipality of Uruará, state Pará, northern Brazil.
|
Ecosystems |
N. areas |
Characteristics |
|
Primary forest – PF |
02 |
Composed of vegetation whose facies is an upper canopy formed mainly by green trees, with crowns that touch each other, creating a dense and enclosed canopy all year round. In this ecosystem no traces of anthropic actions were verified (for example, trails, residues, fires, selective cutting of wood, agricultural activities, among others). Large amounts of litter dominate the ground level, which contributes to the maintenance and fertility of the soil, and serves as food and shelter for several species (Da Silva et al., 2018) |
|
Secondary forest - SF (vegetation with 15 years of regeneration) |
02 |
A secondary forest (or second-growth forest) is a forest or woodland area which has re-grow after abandonment of agriculture. This type of forest arises from natural succession of vegetation, resulting from abandonment after a period of use, a very common action in the Amazon region (Vieira et al., 2008). Biological diversity gradually increases if there are primary remnants to supply seeds. The average height of the trees is over 12 meters and the average diameter is over 14 centimeters. The understory presents usually a dense vegetation, with an intermediate size, greater than 3 meters of height but also with a high amount of litter on the ground (Da Silva et al., 2018). |
|
Incipient Secondary forest– IF (vegetation with five years of regeneration) |
02 |
This type of forest is an embryonic stage of plant succession and consequently presents a more simplified plant composition, with little canopy coverage. These environments have high soil temperature and acidity, little accumulated litter, and a greater presence of several species shrubs, graminoids and forbs (Da Silva, et al., 2018).The average height of trees is no more than four meters and the diameter of the main trees can reach up to eight centimeters This phase typically lasts up to six years and in some cases up to ten years, depending on soil quality and / or seed bank.. |
|
Agroforestry – AF (cocoa plantations) |
02 |
Agroecosystem represented by areas planted with cacao crops (Theobroma cacao L.) in the form of monocultures. Vegetation can reach 5 to 8 m in height, with a closed canopy and a large amount of litter,. Due to the constant cleaning process, the presence and richness of herbaceous and shrubs remains usually low (Santos et al., 2015. |
|
Pasture for extensive livestock – PA |
02 |
Herbaceous vegetation consisting predominantly of exotic grasses, namely of the genus Brachiaria spp, used for extensive livestock farming (e.g., raising cattle). This environment is characterized by high incidence of sunlight soil, soil compaction by cattle, and absence of litter accumulation (Falcão et al., 2015). |
______________________________________________________________________”
Reviewer 1#4:
- Material and methods. 2.2 Data collection and identification: The authors collected material from 480 pitfall traps placed in 10 study plots. Please, add information if the same number of traps was located in each study plot. It would be good to know how the traps were arranged - in line or square design and how the distances between traps were. Please also specify whether the traps were active for 48 hours in each place.
Response 4:
Thanks again for your important question and comment. We have added information regarding the points mentioned. We also, as explained in response 2, corrected the exact number of the total pitfall traps involved in the study research (from 480 to 225). Again, please accept our apologies considering this mistake. Please see explanation for this in response 2.
You may track our changes in the manuscript in the following lines:
Line [134 – 135]: : “In addition, sampling period and intensity, totaling 225 traps…”
Lines [120 – 130]: “Roofed pitfalls of 75 mm diameter and 110 mm deep were filled with preservative liquid made of water and neutral surfactant, to brake the water tension, and coarse salt to conserve individuals captured. Within each ecosystem, three independent areas (minimum distance of 5 km between areas) were chosen and five pitfall traps (minimum distance of 150 meters between pitfalls) were placed for 48 hours, during the three seasons monitored (dry, intermediate and rainy season). Pitfall traps were placed randomly whenever the conditions on the ground where favourable (i.e., no physical our biological barriers such as rocks, trees, etc), always with a minimum distance of 100 m from ecotones, to ensure that most individuals captured were associated with ecosystem monitored.”
_________________________________________________________________________
Reviewer 1#5:
During the study, 14 environmental variables were measured, but nowhere are their values given for particular ecosystems. Such information would show how these environments differ. We can only learn about the differences between study plots from a brief description. So, it would be worth adding the measured values of environmental variables in Supplementary materials.
Response 5:
Thanks again for your comment. New information regarding the ecosystems under study was inserted in a new appendix. Means and associated standard deviation for the environmental variables monitored by ecosystem are depicted in Table A2. - Supplementary Material.
__________________________________________________________________________
Reviewer 1#6:
In Table A3. in supplementary material the environmental variables were listed. Among them were “Number of Plant Species” and “Number of Plants” - in fact, the number of trees, not the number of all plants was taken into account, so it would be good to change the name of this variable.
Response 6:
Thanks again for your pertinent comment and suggestion. We made modifications in these descriptors as suggested. We will be willing to introduce additional modifications, if the reviewer considers necessary.
Table A5. Specification of the methodology used for monitoring the environmental variables associate with the ecosystems studied.
|
Variables |
Specification |
Monitoring Methodology |
|
Temperature |
Celsius (ºC) |
Measured during the traps installation and removal with a portable weather station (model Oregon Scientific WMR200A). |
|
Humidity |
Humidity (%) |
Measured during the traps installation and removal with a portable weather station (model Oregon Scientific WMR200A). |
|
Precipitation |
Precipitation (mm) |
Measured during the traps installation and removal with a portable weather station (model Oregon Scientific WMR200A). |
|
Circumference at Breast Height |
Centimeters (cm) |
Trunk diameter was taken at breast height (1.3 m) for the trees. |
|
Circumference at Ankle Height |
Centimeters (cm) |
The diameter was measured at the ankle height (CAH = 0.1 m) for the shrubs. |
|
Canopy Cover |
Percentage (%) |
Calculated with a convex spherical densiometer (D) Lemmon and assigned the following classes: 0–5%, 6–25%, 26–50%, 51–75%, 76–95% and 96–100% |
|
Richness of Trees |
Tree Species/ m2 |
The number of tree species was counted in an area of 100 m2 (10 x 10 m) in the vicinity of each pitfall trap. |
|
Abundance of Trees |
Number of Trees/ m2 |
The number of tree was counted in an area of 100 m2 (10 x 10 m) in the vicinity of each pitfall trap. |
|
Richness of Shrubs |
Shrubs Species / m2 |
The number of shrubs species was counted in an area of 100 m2 (10 x 10 m) in the vicinity of each pitfall trap. |
|
Abundance of Shrubs |
Number of Shrubs/ m2 |
The number of shrubs was counted in an area of 100 m2 (10 x 10 m) in the vicinity of each pitfall trap. |
|
Percentage of Exposed Soil |
Percentage (%)/ m2 |
The percentage exposed soil in each quadrant was estimated in different percentage classes (0-5, 6-25, 26-50, 51-75, 76-95, 96-100%) |
|
Percentage of Green (vegetation) Cover |
Percentage (%)/ m2 |
The percentage green cover (vegetation up to 1 m height) in each quadrant was estimated in different percentage classes (0-5, 6-25, 26-50, 51-75, 76-95, 96-100%) |
|
Percentages of Leaf Litter Cover |
Percentage (%) |
The percentage of litter in each quadrant was estimated in different percentage classes (0-5, 6-25, 26-50, 51-75, 76-95, 96-100%). |
|
Height of Leaf Litter |
Centimeters (cm) |
Litter height was measured using a ruler at five points inside the square (near each corner and in the center) |
Adapted Da Silva and Hernández (2016)
Reviewer 1#7:
- Results: In 480 traps, only 647 spiders were caught. It seems that this is a small material, especially since the research was carried out in 3 periods. On average, there are 1.35 individuals per trap. It would be good to describe what the variance was. There were probably some traps that a lot of spiders were caught, and there were ones where no individuals were collected. It would be good to write a few sentences about it.
In the Supplementary material, it would be worth mentioning the number of captured individuals in particular study periods, which would also give a better picture of the usefulness of the method used for material collection.
Response 7:
This comment was of extremely importance for us, as discussed in the reply to the question number 4, i.e. the number of pitfall traps, and enabled the detection of a mistake that was already corrected (please check answer to question number 4).
In this sense, we have corrected the sentence and, following the reviewer suggestion, created a new table in appendix that indicates the total number of individuals captured by ecosystem, their mean and standard deviation (i.e., PF, SF, IF, AF and PA). Additionally, individuals were grouped within recognizable taxonomic units (RTU), previously to our subsequent high-level analysis. Please find the table A4 in Supplementary material with this information and the changes made in the text. We are, of course willing to add more information if the referee considers incomplete.
Lines [208-214]:
“A total of 809 individuals were collected, subsequently grouped within recognizable taxonomic units (RTU) and recognizable functional units (RFU). Most individuals were captured in Agroforest (AF) (270 individuals, 18 RTU and 6 RFU) followed by Primary Forest PF (187 individuals, 19 RTU and 7 RFU), Secondary Forest (SF) (173 individuals, 19 RTU and 7 RFU), Incipient Forest (IF) (133 individuals, 16 RTU and 8 RFU) and Pasture (PA) (46 individuals, 15 RTU and 9 RFU) (Table A12 - Supplementary material and Table A2– Supplementary material).”
___________________________________________________________________________
Reviewer 1#8:
- Discussion: In chapter 4.5 Methodological advantages and drawbacks, it would be worth discussing the limitations of the method used, i.e. the fact that the research was conducted with a small number of study plots (there were only two repetitions in each category) and why not so many spiders were collected.
Lines [255 – 462]:
“Our preliminary results demonstrate that high-level indicators reflect gradients of disturbance and consequently, might be considered ecological indicators of the status of neotropical forest landscapes. Anyway, it is likely that the activity of more elusive spiders and species from other strata was probably not detected by pitfalls. Monitoring was accomplished during limited periods, which might not be representative of full activity patterns of several species. Moreover, we also believe that increasing sampling effort and adding more locations should be considered in order to check the feasibility of using our high-level framework as a guideline to monitor the status of neotropical landscapes.”
________________________________________________________
Other minor changes were introduced, namely new references, the correction of grammatical errors and small changes in some expressions. We hope that it fulfils the reviewers and editor expectations and, after this set of corrections, the manuscript might be now considered suitable for publication in Land.

Reviewer 2 Report
Dear authors,
I found your study interesting for publication, as it links functional diversity in tropical forests to disturbance.
However, as you will see in my comments, I feel that several minor aspects should be improved. Namely, I have several questions regarding the sampling design and the environmental variables, which were included in your analysis. I included my comments directly in the pdf. Regarding the supplementary material:
Table A3. Can you please elaborate on how the variables temperature, humidity and precipitation were aggregated for analysis, e.g. did you measure the variables on a daily/hourly basis?
Best regards

Author Response
Dear reviewer, before our replies and explanations of the changes made in order to fulfil your comments and suggestions, we want to express our gratitude for the feedbacks and inputs made to our previous version of the manuscript. We believe that the comments and suggestions enabled a significant improvement of our work. Anyway, if the referee fells that some of the changes and comments are incomplete and/or deserve more information, we will be willing to modify in accordance with additional suggestions.
We have separated below the referees’ comments, highlighted in bold from the authors’ responses, which are depicted in plain text. In addition, we have pasted the changes made in the manuscript and in the supplementary materials using “” and italics, whenever possible.
Response to reviewers’ comments
Reviewers' comments:
Reviewer 2#1:
“If correlated, is the response of these indexes informative, i.e. easy to holistically understand and explain, following a somehow linear trend? “
Reviewer comment: Why would you expect a linear trend?
Response 1:
We agree with the referee that this expression was not particularly clear for the reader. By linear trend we were thinking in a direct cause - effect relationship between pressure and response. As an example, a small anthropogenic disturbance of a native forest implies a minor reduction in biodiversity while significant anthropogenic disturbance would produce a major reduction in biodiversity. Nevertheless, we agree that the sentence might be misunderstood and that correlation maybe not linear, so we decided to eliminate from the sentence “following a somehow linear trend?”.
Therefore, the actual manuscript version has the following sentence:
Lines [93 – 94]: “If correlated, is the response of these indexes informative, i.e. easy to holistically understand, follow and explain?”.
____________________________________________________________________
Reviewer 2#2:
Did you use the exact same locations for pitfall traps during the different periods?
Response 2:
Yes, we used the same locations for pitfall traps during the different periods. If the reviewer considers this information determinant, we will be willing to add it to the text.
______________________________________________________________________
Reviewer 2#3:
3.a - How many per ecosystem & in total?
3.b - What was the distance between traps?
3.c - Were those placed in a specific pattern?
Response 3:
3.a - How many per ecosystem & in total?
We acknowledge the reviewer comments, that we believe most pertinent to increase the information associated to the material and methods.
3.a. Our analyses were performed taking as sample the number of individuals collected by pitfall. For each ecosystem studied (PF, SF, IF, AF and PA), three independent areas were selected (minimum distance of 5km from each other) and five pitfall traps (located with a minimum distance of 150 meters from each other) were placed for 48 hours during the considered seasons (Dry, Intermediate and Rainy). In this way, 225 traps were monitored (225 samples = 5 traps x 3 areas x 3 seasons x 5 ecosystems), each one corresponding to a pitfall trap. Biological data was organized per species or morphospecies or genus (whenever it wasn´t possible to identify to species level). Nevertheless, and to test our high-level hypothesis, individuals were grouped within spider family abundance (HLDa), spider family richness (HLDr) and guilds abundance (HLF). The reason for choosing this approach was because we wanted to test metrics for that we believe useful in regions undersampled and with many undescribed (or poorly known and difficult to be identified) species. In fact, for a high number of individuals captured, taxonomic identification to the species level was particularly difficult (no guides and/or lack of regional taxonomists), but also considering the stochastic variation in species’ captures, linked with unknown variables that could introduce more “noise” to our analysis then clarification.
Also, during the revision of this issue, we realized that we mentioned a wrong number of traps used (480 pitfall traps), when in fact, and as explained above, it was 225. We are deeply sorry for this confusion, and we apologize for this inaccuracy. This confusion aroused because during the period of data collection, other arthropods were also being monitored in the region (e.g. ground beetles, ants, etc) but with a different protocol that include baited traps. We would like also to inform the referee that this mistake had no implications in the analysis made and results obtain. We have corrected in the text the total number of pitfall traps to the actual number. (225). Hence, taking in account reviewer comments we have made the following changes in the manuscript text (we will be willing to add more information if asked by the referee):
Lines [130-133]. In addition, sampling period and intensity, totalling 225 traps, was outlined to link assemblage patterns with ongoing changes, without hankering a full community description …”
We also have added some new information relative to sampling method. Please find it iun the sections bellow of the manuscript new version:
Lines [123 – 126]:
“Within each ecosystem, three independent areas (minimum distance of 5km between areas) were chosen and five pitfall traps (minimum distance of 150 meters between pitfalls) were placed for 48 hours during the three seasons monitored (dry, intermediate and rainy season).”
Lines [166 – 168]:
“HLD was underpinned on RTU spider families’ abundance (HLDa) and RTU spider families’ richness (HLDr) while HLF was supported on RFU sharing ecological traits (spider guilds).”
_____________________________________________________________________________
3.b and 3.c - What was the distance between traps? Were those placed in a specific pattern?
Pitfall traps were placed 150 m from each other and 100 m from ecotones.
We have added this information to the manuscript. Please find actual sentence bellow and in the new revised version of the manuscript.
Lines [120-130]: “Roofed pitfalls of 75 mm diameter and 110 mm deep were filled with preservative liquid made of water and neutral surfactant, to brake the water tension, and coarse salt to conserve individuals captured. Within each ecosystem, three independent areas (minimum distance of 5 km between areas) were chosen and five pitfall traps (minimum distance of 150 meters between pitfalls) were placed for 48 hours, during the three seasons monitored (dry, intermediate and rainy season) [9]. Pitfall traps were placed randomly whenever the conditions on the ground where favourable (i.e., no physical our biological barriers such as rocks, trees, etc), always with a minimum distance of 100 m from ecotones, to ensure that most individuals captured were associated with ecosystem monitored.”
___________________________________________________________________________
Reviewer 2#4:
What kind of preservative did you use?
Response 4:
The preservative used in pitfalls was made of water plus surfactant to brake the water tension and coarse salt to conserve the individuals captured.
This information is now depicted in the main manuscript:
Lines [120-130]: “Roofed pitfalls of 75 mm diameter and 110 mm deep were filled with preservative liquid made of water and neutral surfactant, to brake the water tension, and coarse salt to conserve individuals captured. Within each ecosystem, three independent areas (minimum distance of 5 km between areas) were chosen and five pitfall traps (minimum distance of 150 meters between pitfalls) were placed for 48 hours, during the three seasons monitored (dry, intermediate and rainy season). Pitfall traps were placed randomly whenever the conditions on the ground where favourable (i.e., no physical our biological barriers such as rocks, trees, etc), always with a minimum distance of 100 m from ecotones, to ensure that most individuals captured were associated with ecosystem monitored.”
___________________________________________________________________________
Reviewer 2#5:
“In addition, 110 sampling period and intensity, totaling 480 traps,…”
Reviewer comment: This needs to be more specific.
Response 5:
Please find explanation to this comment in the response to comment 3.a. If the reviewer still finds that is should be improved, we will follow further suggestions.
Reviewer 2#5:
“…probably related with the relevant taxonomic and functional diversity encompassed by the ecosystems studied (Figure 2).”
Reviewer comment: This should be part of the discussion section
Response 5:
We have followed reviewer observation and the sentence is now included in the discussion section (4.1).
Lines [344-345]: “Incomplete rarefaction curves detected for RTU and RFU are probably related with the relevant taxonomic and functional diversity encompassed by the ecosystems studied”
____________________________________________________________________-
Reviewer 2#6:
The reference cited here is about ants, I find it difficult to apply the outcomes of this study on spiders. Additionally, this study refers to results of pitfall traps at species level. I don't think this refernce is is supproting your statement.
Response 6: We agree with the comments of the reviewer and we have substituted by new references of other studies using arthropods within Neotropical ecosystems and high-level analysis.
Lines [227 – 228]:” However, this level of sampling completeness is likely to be sufficient to draw conclusions about the usefulness of HLD and HLF as indicators of anthropogenic impacts [58-59].”
____________________________________________________________________-
Reviewer 2#7:
Please consider a larger font size.
Reviewer comments were applied to figures 2(a) and 2(b).
Response 7: Many thanks for another important suggestion. We made the changes in the font size associated to the figures 2(a) and 2(b) as asked by the reviewer.
Figure 2(a)
-
Figure 2(b)
Reviewer 2#8:
The resolution here is insufficient.
Reviewer comments was applied to figures 4(a) and 4(b).
Response 8: Again, many thanks for helping with these important details. We have changed the output of the graphical information. Point symbols with different colors are associated with each ecosystem.
Figure 4(a)
Figure 4(b)
Reviewer 2#9:
This is note really intuitive to read. Could you please consider a table in addition to the figure.
Reviewer comments were applied to section 3.3. Comparing Taxonomic β-diversity indicators with Functional β-diversity indicators, figures 5(a) and 5 (b).
Response 9: We agree with the referee and two new Tables were added. Table A9 depicting High-level diversity (HLD) dissimilarities and Table A10. Depicting High-level diversity (HLD) dissimilarities, both in to Supplementary Materials.
Reviewer 2#10:
Table A3. Can you please elaborate on how the variables temperature, humidity and precipitation were aggregated for analysis, e.g. did you measure the variables on a daily/hourly basis?
Response 10:
Temperature, humidity and precipitation were measured during 48 hours (an equivalent period to the pitfall traps activation) with a local portable weather station () and linked with online wheatear platforms to check for consistency.Afterwards, these values were aggregated to daily averages.
A new table was introduced and cited in the main manuscript (in the supplementary materials), with the information asked by the reviewer. Also, the text above was introduced above the new table to add understanding what was done.
“Temperature, humidity and precipitation were measured during 48 hours (an equivalent period to the pitfall traps activation) with a local portable weather station () and linked with online wheatear platforms to check for consistency.Afterwards, these values were aggregated to daily averages.”
________________________________________
Other minor changes were introduced, namely new references, the correction of grammatical errors and small changes in some expressions. We hope that it fulfils the reviewers and editor expectations and, after this set of corrections, the manuscript might be now considered suitable for publication in Land.
